E-Volve: understanding the impact of mutations in SARS-CoV-2 variants spike protein on antibodies and ACE2 affinity through patterns of chemical interactions at protein interfaces

Dos Santos Vitor Pimentel 1
Rodrigues André 1
Dutra Gabriel 1
Bastos Luana 1
http://orcid.org/0000-0002-5899-2052 Mariano Diego 1
Mendonça José Gutembergue 2
Lobo Yan Jerônimo Gomes 3
Mendes Eduardo 1
Maia Giovana 1
Machado Karina dos Santos 4
http://orcid.org/0000-0001-7107-5024 Werhli Adriano Velasque 4
Rocha Gerd 2
http://orcid.org/0000-0003-4346-9880 de Lima Leonardo Henrique França 3
http://orcid.org/0000-0001-5190-100X de Melo-Minardi Raquel 1 raquelcm@dcc.ufmg.br
1 Laboratory of Bioinformatics and Systems, Institute of Exact Sciences, Department of Computer Science, Universidade Federal de Minas Gerais , Belo Horizonte, MG , Brazil
2 Laboratory of Quantum and Computational Chemistry, Center of Exact and Natural Sciences, Department of Chemistry, Universidade Federal da Paraíba , João Pessoa, PB , Brazil
3 Laboratory of Molecular Modeling and Bioinformatics, Campus Sete Lagoas, Department of Exact and Biological Sciences, Universidade Federal de São João del-Rei , Sete Lagoas, MG , Brazil
4 Computational Biology Laboratory (ComBi-Lab), Center for Computational Sciences-C3, Universidade Federal do Rio Grande , Rio Grande, RS , Brazil
Naveca Felipe
Electronic publication date: 2022 Mar 22
Publication date: 2022
Volume: 10
Electronic Location ID: e13099
Received 2021 Sep 24; Accepted 2022 Feb 21
Copyright: © 2022 Dos Santos et al.
Copyright year: 2022
Copyright holder: Dos Santos et al.
License: This is an open access article distributed under the terms of the Creative Commons Attribution License, which permits unrestricted use, distribution, reproduction and adaptation in any medium and for any purpose provided that it is properly attributed. For attribution, the original author(s), title, publication source (PeerJ) and either DOI or URL of the article must be cited.
License URL: https://creativecommons.org/licenses/by/4.0/

Keywords: SARS-CoV-2, Antibody, Variant, Vaccine, ACE2, Chemical interactions, Affinity, Web tool

Funding: Coordenação de Aperfeiçoamento de Pessoal de Nível Superior-Brasil 001 (51/2013-23038.004007/2014-82). Laboratório Nacional de Computação Científica–LNCC, Petrópolis/RJ, Brazil Centro Nacional de Processamento de Alto Desempenho em São Paulo Núcleo de Processamento de Alto Desempenho of Universidade Federal do Rio Grande do Norte This study was financed by the Coordenação de Aperfeiçoamento de Pessoal de Nível Superior-Brasil (CAPES)-Finance Code 001 (51/2013-23038.004007/2014-82). Physical structure and computational support were provided by the Laboratório Nacional de Computação Científica–LNCC, Petrópolis/RJ, Brazil (Project “Prospecção e testes in vitro de inibidores de proteínas associadas ao vírus SARS-Cov 2 por meio do uso conjunto de ferramentas de bioinformática, simulação molecular, química quântica e aprendizado de máquina-qcbiocovid19” Supercomputer SDumont); Centro Nacional de Processamento de Alto Desempenho em São Paulo (CENAPAD-SP); Núcleo de Processamento de Alto Desempenho of Universidade Federal do Rio Grande do Norte (NPAD/UFRN). There was no additional external funding received for this study. The funders had no role in study design, data collection and analysis, decision to publish, or preparation of the manuscript.

==============================
Background

The SARS-CoV-2 pandemic reverberated, posing health and social hygiene obstacles throughout the globe. Mutant lineages of the virus have concerned scientists because of convergent amino acid alterations, mainly on the viral spike protein. Studies have shown that mutants have diminished activity of neutralizing antibodies and enhanced affinity with its human cell receptor, the ACE2 protein.

Methods

Hence, for real-time measuring of the impacts caused by variant strains in such complexes, we implemented E-Volve, a tool designed to model a structure with a list of mutations requested by users and return analyses of the variant protein. As a proof of concept, we scrutinized the spike-antibody and spike-ACE2 complexes formed in the variants of concern, B.1.1.7 (Alpha), B.1.351 (Beta), and P.1 (Gamma), by using contact maps depicting the interactions made amid them, along with heat maps to quantify these major interactions.

Results

The results found in this study depict the highly frequent interface changes made by the entire set of mutations, mainly conducted by N501Y and E484K. In the spike-Antibody complex, we have noticed alterations concerning electrostatic surface complementarity, breaching essential sites in the P17 and BD-368-2 antibodies. Alongside, the spike-ACE2 complex has presented new hydrophobic bonds.

Discussion

Molecular dynamics simulations followed by Poisson-Boltzmann calculations corroborate the higher complementarity to the receptor and lower to the antibodies for the K417T/E484K/N501Y (Gamma) mutant compared to the wild-type strain, as pointed by E-Volve, as well as an intensification of this effect by changes at the protein conformational equilibrium in solution. A local disorder of the loop α1′/β1′, as well its possible effects on the affinity to the BD-368-2 antibody were also incorporated to the final conclusions after this analysis. Moreover, E-Volve can depict the main alterations in important biological structures, as shown in the SARS-CoV-2 complexes, marking a major step in the real-time tracking of the virus mutant lineages. E-Volve is available at http://bioinfo.dcc.ufmg.br/evolve.

Introduction

With the pandemic of coronavirus disease 2019 (COVID-19) and the intense study of its pathogen, Severe Acute Respiratory Syndrome Coronavirus 2 (SARS-CoV-2), an unprecedented volume of sequences and structures have been generated (Hu et al., 2021). SARS-CoV-2 is a sarbecovirus from the betacoronavirus genus with a single-stranded, positive-sense ribonucleic acid (RNA). Its genomic RNA contains approximately 30,000 base pairs and poses about 88%, 89% of phylogenetic proximity with SARS-like viruses from bats, described as bat-SL-CoVZC45 and bat-SL-CoVZXC21 (Lai et al., 2020).

Large-scale SARS-CoV-2 genome sequencing is essential for epidemiologic surveillance and control (Rambaut et al., 2020). A particular global apprehension has been the new concerning variants identified in December 2020. Although viruses naturally mutate at high rates, these variants pose additional concerns for two main reasons: (i) they seem to be more transmissible and possibly more infective, and (ii) the extent of the impact on the effectiveness of developed vaccines is not known for sure (Rambaut et al., 2020).

The B.1.1.7 variant (Alpha), first discovered in December 2020 in the UK, was brought to the attention of authorities by the high rate of infection. The Alpha variant fastly spread across dozens of countries and carried 17 mutations, including eight in the spike protein (HV 69-70 deletion, Y144 deletion, N501Y, A570D, P681H, T761I, S982A, and D1118H), which are the basis of three COVID-19 vaccines licensed (Rambaut et al., 2020). B.1.351 (Beta) was identified in South Africa at the end of 2020. The Beta variant has non-synonymous spike mutations such as LAL 242-244 deletion, D80A, D215G, E484K, N501Y, A701V, L18F, R246I, K417N, and D614G (Tegally et al., 2020). P.1 variant (Gamma), detected to circulate in Brazil by December 2020, is probably implicated in the peak of infections faced by Manaus in the Brazilian Amazon. It presents the following mutations: L18F, T20N, P26S, D138Y, R190S, K417T, E484K, N501Y, H655Y, and T1027I (Naveca et al., 2021b, 2021a).

Additionally, Alpha, Beta, and Gamma lineages pose apprehension for the impact they could cause. As in cases of reinfection (Naveca et al., 2021b, 2021a), by the non-synonymous convergent spike mutations N501Y, appearing in all of them, E484K occurring on the Gamma and in the Beta and K417T taking place in the Gamma variant. These cited spike mutations arose in the viral receptor binding site (RBD) and affected the human as studied (Nelson et al., 2021). One of those effects revealed by Nelson and collaborators is the increased affinity between the Receptor-Binding Domain (RBD) and the Angiotensin-Converting Enzyme 2 (ACE2) protein caused by the N501Y and E484K mutations, which also have their potentiality enhanced when possessing the K417N/T modification. Also, these lineages are designated as variants of concern (VOCs), because of their rapid growth and fast predominance as the main lineages of SARS-CoV-2, in each respective region (Galloway et al., 2021). In October of 2020, from the B.1.617 clade, emerged the B.1.617.2 SARS-CoV-2 strain—known as the Delta variant, by the WHO nomenclature, or as 21A, by the Nextstrain nomenclature. First described in India, this strain has become predominant worldwide. It is associated with the third epidemiological wave in the United Kingdom, Brazil, South Africa, The United States, and many other countries (Callaway, 2021).

According to Burki (2021), Moderna and Pfizer vaccine manufacturers confirmed that their vaccines would continue to be effective against Alpha. Also, in relation to Beta, Pfizer was associated with a reduction in neutralizing antibodies. Hence, it is crucial to develop tools that allow us to understand better how each mutation can affect at the molecular level: (i) the interactions with antibodies, and (ii) the affinity between the spike protein and the human receptor Angiotensin-Converting Enzyme 2 (ACE2).

In this study, we propose a web tool called E-Volve to model and analyse new and emerging variants only by knowing the mutations in the strain sequence. The tool enables a real-time understanding of possible impacts of mutant SARS-CoV-2. Also, we obtained the sequences of the other three main variants in circulation from GISAID (Shu & McCauley, 2017) and the structures available in the Protein Data Bank (PDB) of the spike protein in complex with antibody fragments and with the enzyme ACE2. Thus, we scrutinized the interaction patterns on the interfaces of each complex. Furthermore, we show the molecular alterations that occur in the spike protein of the Delta strain, which are responsible for the effects seen so far concerning ACE2 and antibody interactions. We utilized the L452R, P681R, and E484Q key mutations of Delta, to model the variant’s interface against both complexes.

Materials and Methods

Data collection

We gathered the structures for SARS-CoV-2-reactive human antibodies of the COVID-19 resources from the PDB (Protein Data Bank) (Berman et al., 2000). For example, we collected the structure of PDB ID 7KKL that contains SARS-CoV-2 spike in a complex with neutralizing nanobodies and the 6M0J spike protein in a complex with the human ACE2. A complete table containing the PDB IDs and the collection date is available on the Supplemental Materials (Table S1). Furthermore, we intend to update our analyses using E-Volve as new data entries of SARS-CoV-2 are available in the PDB.

3D-structures modelling

For modelling the Gamma and Delta lineages, we used the MODELLER software (Webb & Sali, 2016). The modelling pipeline consisted of three steps: (i) template selection based on sequence similarity (except for point mutations, where the wild structure was used as a template), sequence alignment, comparative modeling, and model validation. We also used the Mutate.py script to apply point amino acid substitution in each spike protein of our database. As for the Alpha and the Beta clades, we applied the mutations directly in the sequence. We used the standard fast full modeling procedure of MODELLER for modeling each SARS-CoV-2 mutated sequence. We chose this procedure because Mutate.py cannot apply amino acid deletions. Furthermore, experimentally resolved structures of these variants can be found in the PDB and were selected based on sequence similarity.

Then, we evaluated the quality of the generated models, being one per mutant, using the MODELLER objective function and best template sequence identity in percentage (Webb & Sali, 2016). We used the models with the smallest objective function, or Discrete Optimized Protein Energy (DOPE), scores (Shen & Sali, 2006) for all structures scrutinized. Furthermore, the 6M0J templates for the Gamma, Alpha, and Beta strains obtained sequence identity of 99.87%, 99.87%, and 99.62%, respectively.

Also, for all the spike-antibody complexes built in this study, we calculated the arithmetic mean for the sequence identity measures and a population standard deviation to ensure the quality of the models. For Gamma, Alpha, and Beta lineages, we have found a mean sequence identity of 97.71%, 94.20%, and 98.97%, with standard deviation measurements being 14.69, 22.78, and 7.38, respectively. The generated models posed a sequence identity more significant than 90% of coverage with high DOPE scores.

Algorithm/Method

The E-Volve’s strategy is based on detecting conserved contacts when comparing mutant and wild protein structures. For example, a single-point mutation can change a type of interaction between two residues that perform non-covalent chemical interactions, such as changing an attractive polar interaction by a repulsive. This could lead to severe changes in the protein structure, mobility, and interactions with other proteins and ligands. Hence, comparing the differences in the contact types caused by mutations can help to understand their impact on the molecular mechanisms.

We used a modified version of the VTR algorithm (Pimentel et al., 2021) to calculate and compare original viral spikes and their respective mutant model interactions. This method detects all contacts into a protein pair. Then, VTR uses the TM-align structure superposition method (Zhang & Skolnick, 2005) to perform alignment between each mutant structure against the wild-type. Lastly, VTR detects equivalent interacting residue pairs in each complex. This step is called contact matching.

The VTR algorithm utilizes determined cutoff distances for each contact type based on previous studies (Fassio et al., 2019; Silva et al., 2019; Barroso et al., 2020). A hydrophobic interaction has a minimum distance of 2 Å and a max distance of 4.5 Å. An attractive/repulsive contact has a minimum distance of 2 Å and a max distance of 6 Å. Hydrogen bonds are determined by ≤3.9 Å. The aromatic stacking interaction is made between 2 and 6 Å. Lastly, disulfide bonds pose a minimum distance of 1.5 Å and a max distance of 2.8 Å.

Database

We build a database that contains the modelled mutated structures and the contact comparison between the original and the mutated structures. This database was constructed using the structures gathered from PDB, the method developed for modelling, and E-Volve’s strategy. The database is divided into eight sections, grouped for each lineage mutation applied to spike-ACE2 complexes and one for each lineage mutation applied to spike-antibody complexes. The database is available at http://bioinfo.dcc.ufmg.br/vtr/covid_home.

Web tool

Based on the method described above, we constructed E-Volve, a web tool that allows the user to analyse any protein complex and its mutants. E-Volve, receives from the user a protein complex in PDB format, a list of mutations and pairs of chains to be analyzed. E-Volve creates a model of the mutated protein and aligns the contacts between the chain pairs of the given template with the mutated model generated in the output. The web tool can be accessed through an interactive web server implemented using the CodeIgniter PHP framework. In addition, protein three-dimensional structure visualizations were generated using the 3Dmol.js library (Rego & Koes, 2015).

Contact maps and heat maps

We created contact maps to analyze losses and gains of meaningful interactions that emerged in the SARS-CoV-2 VOCs studied in contrast with the referential wild-type. The graphs generated assemble the location of each mutation, the site in contact with the examined antibody and the chemical interaction between them. Furthermore, we superposed all the contact maps to visualize the most convergent interactions made amidst all the sampled antibodies.

Additionally, we used the Plotly graphing library (Sievert, 2020) to generate heat maps to evaluate the frequency in each contact’s occurrence and the importance of each residue in the interface affinity. These graphs are a superposition of all contact maps of spike-antibody and spike-ACE2 that we built in our database. The color gradient varies from red to blue, representing a small number of contacts and numerous contacts, respectively. E-Volve also highlights mutations provided by the user, showing how these mutations can affect the protein interactions with the antibodies or with the ACE2. We consider the heat maps complementary to the contact maps due to their potential capacity of measuring intensity. For example, in the x-axis of the proposed heat maps, we can present the protein spike residues, and in the y-axis, the residues from all the antibodies or the ACE2. We built heat maps for each nature of the interaction (e.g., attractive, repulsive, aromatic stacking, hydrogen bonds, hydrophobic).

Molecular dynamics simulations and surface electrostatic potential calculations

To evaluate the E-Volve results, we performed molecular dynamics (MD) simulations followed by continuum electrostatics calculations for the protein surface potential of the wt and Gamma RBDs. We evaluated how much the three amino acid substitutions from the spike wild-type (wt) to the Gamma lineage at the receptor-binding domain (RBD) reflect on the intrinsic conformational equilibrium and average electrostatic potential at the principal binding site of this protein.

System preparation and MD simulations

The crystallographic structure of the wild-type SARS-CoV-2 spike glycoprotein was collected from the Protein Data Bank (PDB) 6M0J. The residues 333 to 527 that roughly correspond to the receptor-binding domain (RBD) were selected using PyMOL (The PyMOL Molecular Graphics System, Version 2.0 Schrödinger, LLC). A Gamma lineage RBD (with mutations E484K, N501Y, and K417T) was modelled comparatively using SWISS-MODEL (Waterhouse et al., 2018). Protonation states for all residues in both systems were estimated for the standard physiological blood pH of 7.4 and salinity of 0.15 M using the H++ web server (Anandakrishnan, Aguilar & Onufriev, 2012). The systems were parameterized using the tleap tool from AmberTools21 (Case et al., 2008). The systems were solvated on a cubic box with a minimum border distance of 12 Å, and the water, protein, and glycidic portions of the systems were respectively described with the TIP3P, FF14SB, and GLYCAM-06 AMBER force fields. Na+ and Cl- ions were added in order to reach the physiological salinity of 0.15 M. Afterwards, the minimization, equilibration and proper productive MD simulation steps for both systems were run using NAMD 2.13-verbs-CUDA (Phillips et al., 2005). All simulations were carried out with the NPT ensemble, with the temperature and pressure constantly maintained respectively at 310 K and 1 atm by a Langevin thermostat and Langevin piston. Periodic boundary conditions were established following a 10 Å cutoff for nonbonded interactions, and the long-ranged electrostatic interactions were calculated with the particle mesh Ewald (PME) method.

Prior to the proper productive MD simulation, a multistep equilibration protocol was carried out, following: (a) 500 steps of minimization with harmonic restraints on the protein atoms; (b) 500 steps of minimization without restraints; (c) 300 ps of equilibration with harmonic restraints on the protein atoms; (d) 300 ps of equilibration with harmonic restraints on the backbone atoms; (e) 300 ps of equilibration without restraints; and (f) 1 ns of a pre-productive simulation with reinitialized velocities. Finally, three independent 100 ns productive MD simulations were run for each system (wt and Gamma RBD), maintaining the system general sets as above mentioned, on the SDumont supercomputer, in the Laboratório Nacional de Computação Científica (LNCC). Only the last 50 ns of each trajectory were taken for further analysis.

Trajectory analysis

Per residue root-mean-square fluctuation (RMSF) analyses against the average frame for the Cɑ carbons were carried out using the cpptraj plugin from AmberTools21 (Case et al., 2008). For both systems, two sets of analyses were carried out. First, we aligned the whole RBD considering the least-mobile residues (333–437, 454–456, 492–494 and 509–526). Then, we analysed the fluctuations for the whole domain. Afterwards, we aligned and analyzed only the alpha carbons of the residues corresponding to the receptor-binding motif (RBM) (438 to 508), in order to observe its internal movements (Yi et al., 2020). Cluster analysis of the trajectories resulting from MD was performed using the cpptraj plugin from AmberTools21 (Case et al., 2008), using the RMSD over the RBM region as a metric. The single linkage method for each respective system was applied considering the last 50 ns of their three replicates, each 50 ns amount distributed on 1,563 frames of about 32 ps each frame (totalizing a clustering procedure for a 150 ns ensemble distributed on 4,689 conformers for each system). The procedure was adjusted to recover three representative clusters for each system, their respective average distance, the MD percentual representativity, and the cluster density parameters.

Surface electrostatic calculations by APBS

We have taken the three respective cluster centroid conformations recovered for each system by the above-mentioned clustering procedure to Poisson-Boltzman electrostatic potential surface area (PBSA) by the online versions of PDB2PQR and the Adaptive Poisson-Boltzmann Solver (APBS) tools (Dolinsky et al., 2004, 2007; Jurrus et al., 2018). The respective protonation states and AMBER topologies used at the MD simulations were conserved at the PBSA calculations. Also, the parameters related to the same pH 7.4 and ionic strength of 0.15 M, as well as default internal and external dielectric constants, were considered at the implicit solvent approximations. In order to have a first glimpse about the electrostatic complementarity of representative crystal structures of the ACE2 receptor and the P17 and BD-368-2 antibodies, the respective chains corresponding to the structures 6M0J, 7CWN, and 7CHH were taken to protonation followed by PBSA calculations, respectively at H++, PDB2PQR, and APBS tools at the same pH, ionic strength, and dielectric conditions. In sequence, the centroid conformations of the two major MD clusters for each system were aligned to the RBM region of the RBD spike from these three respective PDBs using the PYMOL software to obtain the respective RBD-ACE2 and RBD-antibody complexes. Consequently, the electrostatic complementarity of the PBSA maps between the RBD and the macromolecular partner was visually analyzed in each case.

Results

Web tool

Here we show E-Volve, a web tool composed of a pipeline to systematically analyze interchain interactions, and thus, evaluate the impact of mutations in protein contacts. We used E-Volve to analyze interaction variations of SARS-CoV-2 spike proteins. E-Volve receives as input a protein file in PDB format or a PDB identifier, a list of desired mutations to be applied on the former file, and a list of chain pairs to evaluate their interaction using a contact comparison strategy (Fig. 1A). To calculate and compare the interface between the original protein contrasted with the mutated protein (Fig. 1B), E-Volve selects the best modeling method by either using the Modeller standard modeling or the Modeller mutate.py script. Afterward, it generates a set of mutated protein files (Fig. 1C).

Figure 1 E-Volve workflow.

(A) E-Volve receives as input a PDB file or a four characters PDB ID code. Also, a list of mutations must be informed during input submission. (B-top) E-Volve detects intra-chain and inter-chain interactions in the wild and mutant proteins. (C) E-Volve uses MODELLER software to perform comparative modeling of mutants using the wild protein as a template. (B-bottom) E-Volve uses TM-align to perform pairwise structural alignment between the wild protein and each mutant. Then, E-Volve uses the VTR algorithm to calculate contact matches, i.e., detects interacting residue pairs in analogous positions for wild and mutant structures. (D) E-Volve shows interactive visualizations of contact conservation, such as heat maps, contact maps, equivalent contact tables, 3D visualization of the protein structures superimposed, and a statistical summary.

E-Volve uses the VTR algorithm. Its output consists of a summary page with helpful information about the execution (Fig. 1D-left). It also contains a button for downloading the wild and mutant proteins, and a list of interaction analyses. E-Volve also shows a heat map with the more impacted interactions by the mutations. Additionally, E-Volve allows users to click on a “show details” button. This button leads them to a page that allows exploring interactions through 3D-structure visualizations using the 3Dmol JavaScript plugin, in addition to interactive contact maps and tables (Fig. 1D-right).

SARS-CoV-2 case study findings

To evaluate E-Volve, we analyzed the inter-chain interactions between Sars-CoV-2 spike RBD protein in complex with neutralizing antibodies. We also analyzed the Sars-CoV-2 spike RBD protein bound to the cell receptor ACE2. Our main objective is to evaluate the differences in 3D structures among wild spike proteins and mutants (Alpha, Beta, and Gamma). We also aim to verify the impact in their interactions with ACE2 and antibodies. Briefly, the E-Volve results highlight the importance of several residues, particularly 417, 484, and 501.

We used several structures of the spike in different conformations (a complete PDB ID list is available at Supplemental Material). For example, we aligned with the E chain from 6M0J (RBD), A chain from 6VXX (spike in the closed state), and C chain from 7DK3 (spike in the open state), and we define that a convergency exists if the TM-Score was more significant than 0.5. This cutoff is based on the Xu & Zhang (2010) study. Thus, to acquire a TM-score of no less than 0.5, we should consider at least 1.8 million random protein pairs (p-value of 5.5 × 10−7). E-Volve uses this information to build a heat map that shows how the mutations can affect essential regions of the protein using a previously generated database. This also shows how the modelled mutations could impact the interactions between the Sars-CoV-2 spike and ACE2 human protein and between spike and some antibodies.

Figure 2A demonstrates the superposition of all generated contact maps: the interaction pairs among the viral spike protein with all the complexed antibodies. In this graph, each dot represents an inter-residue interaction. Different colors indicate the contact type, which can be yellow (disulfide bond), magenta (attractive interaction), cyan (repulsive interaction), purple (hydrogen bonds), green (aromatic stacking), or brown (hydrophobic). This graph can help us detect amino acid substitutions that promote a subtle contact change. For example, in Fig. 1, we highlighted the E484K mutant, which causes a change in the interaction from attractive to repulsive. Furthermore, we marked the known mutation points for the Gamma lineage with asterisks. Thus, this figure reflects that most of the spike-antibody complex interface is sensitive to RBD mutations once the primary connection in this complex occurs in this site, usually demarcated from the 333 to the 527 spike residue.

Figure 2 Spike-antibody contact and heat maps.

(A) Superposition of the contact maps between spike protein and antibody fragments. Spike residues are shown in the X-axis. Antibody residues are shown in Y-axis. We highlighted position 484: the E484K mutation changes the attractive interaction to a repulsive. Mutations described in the literature for spike protein are shown as black asterisks. Interactions: disulfide bonds (yellow), attractive (magenta), repulsive (cyan), hydrogen bond (purple), aromatic stacking (green), and hydrophobic (orange). Available at http://bioinfo.dcc.ufmg.br/evolve/results. (B) Heat map depicting the frequencies of attractive spike-antibody interactions. The bluer, the more interactions were detected between both residues in different structures.

Figure 2 allows a comparison between the contact maps (Fig. 2A) and the superposed heat map (Fig. 2B) of attractive ionic contacts established between the spike protein and the antibody fragments for the Sars-CoV-2 data set used. Notice that several positive residues can interact with the spike’s glutamate E484. According to our large-scale contact detection, follows the list of positive residues that interact with E484: K30, H35 (in 62 complexes), R50, R56, K65, R96 (in 134 complexes), R98 (in 78 complexes), H99 (in 26 complexes), R100 (in 20 complexes), R102 (in 47 complexes), R107 and R112. Figure 3A presents P17, and positively charged residues are depicted in spheres.

Figure 3 Changes in protein structure caused by amino acid substitutions (Gamma lineage, P17 antibody, and ACE2).

(A) Three attractive interactions lost in E484K. In red, the P17 antibody fragment. In orange, the spike chain. In orange spheres: E484; purple: H35 and H99; and cyan: R98. (B) New contacts emerged in the Gamma lineage depicting the interaction between the ACE2 in green and the viral spike protein in blue, with residues labeled and the 6M0J mutant model superposed with the original 6M0J ACE2 complex. Figure generated using PyMOL.

Analyzing the contacts between the first lineage’s viral spike protein and the sampled antibodies, we assert a difference in the interaction for the 484 residue alteration present on the Beta and Gamma lineages. The E484K mutation jeopardized a key contact for the antibody neutralization, which could partially explain the diminishing, without terminating, immune response as studied by Jangra et al. (2021). Forbye the Alpha lineage, we show the posing of the E484K and K417N, present in the Alpha and the Beta strains, taking shape into the graphs as a constant repulsive interaction amongst the spike-antibody complexes scrutinized.

Alongside the breach in the spike-antibody complex caused by the E484K transformation, the effects also go further when evaluating the spike-ACE2 interaction. This positively charged lysine increases binding affinity within the cited human cell receptor, expressing the danger of reinfection already seen by Nonaka et al. (2021). Beyond this single point mutation, the entire set of alterations on the viral RBD such as E484K, N501Y, and K417N stabilize the VOCs as epidemiologically dominant. Thus, they can endanger the potential of mRNA vaccines such as Moderna and Pfizer without terminating their immunogenic competence, as foreseen by Wang et al. (2021a, 2021b).

Moreover, in the SARS-CoV-2 RBD, the N501Y mutation is convergent in the three VOCs here evaluated. It has posed epidemiological distress for its association with an increased affinity with the human ACE2 (Nelson et al., 2021). In our study, we map and measure the frequency of this interaction. We show that most of the sampled SARS-CoV-2 PDB receptors take a sight into the heat maps as highly predominant in an amplified affinity interface within the spike-ACE2 complex.

Discussion

The results provided by the identification and matching of contacts and the molecular docking, amongst the complexes, rebound a persistent breach on the interactivity between the viral RBD and the significant amount of sampled PDB antibodies, chiefly guided by the E484K alteration. The spike-ACE2 configuration presents constant affinity-based contacts due to the N501Y mutation on the VOCs examined, clarifying results such as shown by Ali, Kasry & Amin (2021), elucidating this vehement binding affinity present in this complex.

Molecular dynamics (MD) simulation followed by electrostatics calculations by Poisson-Boltzmann surface analysis (PBSA) confirmates the major E-volve findings at single structure analysis. That is, the most representative conformations adopted by the receptor-binding domain (RBD) of the Gamma lineage present higher stereochemical complementarity in its receptor binding motif (RBM) to ACE2 and lower to antibodies. In addition, the MD and PBSA results suggest that the three substitutions in Gamma lineage (K417T, E484K, N501Y) can still intensify this higher complementarity of the RBM to the receptor and lower to the antibodies by shifting the RBD conformational equilibrium. This favors conformations more complementary to ACE2, less complementary to antibodies, and disfavors conformations with different behaviour. Furthermore, the MD results suggest an additional penalty for the binding of the loop α1′/β1′ at the Gamma lineage, due its particular conformation and dynamics. The results suggest that this penalty can be considerably higher at the antibody complexation (overall BD-368-2), which may be a further acquired mechanism of antibody resistance for the P1 RBD. All these additional findings suggest a binding mechanism with the participation of conformational selection, a phenomenon both already widely pointed as a key factor for protein-protein and biomolecular recognition, as specifically suggested for spike interactions (Pallara et al., 2016; Ahamad, Hema & Gupta, 2021; Rocha et al., 2021). Also, for epitope interactions with antibodies and other immune system receptors, the literature points to the significant relevance of protein flexibility and conformational selection (Fieser et al., 1987; Stejskal et al., 2020). In future studies (already being conducted on our group), we aim to better explore these dynamics features by considering a higher number of mutants and using more robust MD analysis and free energy calculations.

Moreover, the results show a correlation involving a more frequent interaction on the spike-antibody complexes presenting the K417N mutation, which has taken the view of scientists for a more stabilized interface, such as shown by Starr et al. (2021) and Tian et al. (2021). The pose of the K417N alteration frequently jeopardizes the monoclonal antibody serum activity for the SARS-CoV-2 infections, being present on the Beta and Gamma strains alongside the E484K mutation (Naveca et al., 2021b, 2021a). Again, our MD analysis points the local packing alterations with a substitution that similarly removes the charge and shortens the side chain at the position 417 (i.e., the K417T substitution from wt to P1) as an important factor to shift the RBM dynamics and, eventually, stabilize a less antibody complementary conformation.

These results may contribute to explain the herein lineage’s increased capacity of infectivity (Nelson et al., 2021), associated with more affinity between the viral RBD and the human ACE2 (Ali, Kasry & Amin, 2021; Tian et al., 2021). This occurrence may lead to augmentation for long-term transmissibility of the SARS-CoV-2 VOCs, for being impactful on the capacity of viral replication, and further dominating more basal strains as visible on real-time tracking by Nextstrain (Hadfield et al., 2018).

Our research presents data explaining the convalescent serum neutralization escapability amongst lineages presenting the E484K mutation, alongside the N417K/T alteration, on the viral RBD. However, because of the behaviour presented by Beta and Gamma variants, further investigation around immunogenic suppression of the infection in these lineages is still necessary (Bettini & Locci, 2021).

Spike-ACE2 complex

We found an altered interface between the overall viral spike and the human ACE2 structures based on the 6M0J PDB spike/ACE2 complex. The effects of mutations in Alpha, Beta, and Gamma lineages were classified according to the detected contactual difference. Beta and Gamma have several altered interactions between the spike’s 484 residue and ACE’s 31 residue, mainly due to the E484K mutation, which accounts for changing from an attractive to a repulsive interaction in this site. Notwithstanding, the Gamma lineage has shown a new attractive bond involving this location in the K484/E35 bond.

Additionally, the N501Y convergent mutation in these three strains studied has shown a crucial interfacial remolding of the spike-ACE2 complex. This is denoted by the constantY501/K353 hydrophobic interaction. This interface can increase the final binding affinity between the viral RBD and the ACE2 in the studied lineages. Moreover, the Gamma strain has also shown a new contact, the Y501/Y41 bond, leading to another recurrent hydrophobic interaction in the spike-ACE2 complex and impacting furthermore in this lineage’s general interface for the human ACE2 and the viral RBD.

Table 1 depicts the interaction differences in the 6M0J complex. It demonstrates the viral spike residue positions intercalated between the wild and the mutated amino acids. In addition, the spike-altered residues appear in a complex with the human ACE2 fragments to exhibit the chemical changes in the contacts and the appearance of new highly frequent interactions. Figure 3B depicts the frequent contacts that emerged in the P1 lineage, denoting the E484K mutation in hydrogen bonds with the K31 and E35 human ACE2 residues, alongside the N501Y alteration in hydrophobic interactions with the K353 and Y41 residues.

Table 1 New interactions that appeared on the interface spike-ACE2 complex for the P.1, B.1.351, and B.1.1.7.

PDB ID	Mutation (interacting with)	Linage (interaction type)	
P.1	B.1.351	B.1.1.7	
6M0J	E484K (K31)	Attractive → Repulsive	Attractive → Repulsive	–	
E484K (E35)	Attractive	–	–	
N501Y (Y41)	Aromatic stacking or Hydrophobic	Hydrogen bonds	Hydrogen bonds	
N501Y (K353)	Hydrophobic	Hydrophobic	Hydrophobic	
Note:

The arrow indicates that the mutation changes the interaction type.

Spike-antibodies complex

The results found in major PDB complexes, depicting neutralizing antibodies interacting with the viral spike protein (7CWU, 7CWN, 7CWS, and 7CHH), allow us to conceive the difference between SARS-CoV-2 mutant lineages and wild-type strains. We choose to analyze such models for their complex with the P17 and the BD-368-2 antibodies. These antibodies cover the main spike altered sites of the VOCs, positions 484 and 501. Thus, we have analyzed mutations that give rise to the Alpha, Beta, and Gamma lineages and their neutralization by antibodies covering the cited positions.

In the Gamma strain, the P17 neutralizing antibody present in the 7CWU and 7CWN demonstrated significant differences because of the transition from former hydrogen bonds and attractive interactions to a repulsive interface, evaluated in the H35, R96, R98, and R99 residues. Furthermore, the BD-368-2 neutralizing antibody-associated within the 7CHH complex also reveals a changing interface from attractive interactions and hydrogen bonds to repulsive linkages present in the R100 and R102 residues. Such antibody evasion has been denoted by Dejnirattisai et al. (2021). The 7WCS complex is not affected by such mutations in any of the studied lineages once its connected antibodies are not interacting with the viral RBD.

Table 2 denotes the interaction between the mutated spike proteins and the RBD neutralizing antibodies sites. Positions H35, R96, R98, and H99 constitute residues from the P17 antibody fragment, whereas R100 and R102 are amino acids from the BD-368-2 neutralizing antibody. The table demonstrates the main altered contacts between these interfaces alongside the respective chains in which the modification occurs. Figure 4 demonstrates the former and new interactions between the Gamma strain and the human antibodies focusing on the cited residues.

Table 2 Differences in the interaction between the wild-type viral spike protein and the mutant P.1, B.1.351, and B.1.1.7 lineages with the main neutralizing antibodies sites for 7CWU, 7CWN, and 7CHH.

PDB ID	Residue (interacting with)	Linage (interaction type)	
P.1	B.1.351	B.1.1.7	
7CWU	H35 (P17)	(B-H) Attractive/Hydrogen bond → Repulsive.	–	–	
R96 (P17)	(B-L) Attractive → Repulsive	(A-S; C-R; B-L) Attractive → Hydrogen bonds	–	
R98 (P17)	(C-M) Attractive → Repulsive	–	–	
H99 (P17)	(B-H) Attractive → Repulsive/Hydrogen bond	–	–	
R100 (BD-368-2)	–	–	–	
R102 (BD-368-2)	–	–	–	
7CWN	H35 (P17)	(A-G; C-L) Attractive/Hydrogen bond → Repulsive (B-K) Attractive/Hydrogen bond → Repulsive Hydrogen bond	–	–	
R96 (P17)	(A-F; B-I; (C-J) Attractive → Repulsive	(A-F; C-J; B-I) Attractive → Hydrogen bonds	–	
R98 (P17)	(C-L; A-G; B-K) Attractive → Repulsive	–	–	
H99 (P17)	(C-L; A-G; B-K) Attractive → Repulsive (C-L; B-K) Attractive → Repulsive/Hydrogen bond	–	–	
R100 (BD-368-2)	–	–	–	
R102 (BD-368-2)	–	–	–	
7CHH	H35 (P17)	–	–	–	
R96 (P17)	–	–	–	
R98 (P17)	–	–	–	
H99 (P17)	–	–	–	
R100 (BD-368-2)	(A-D; C-J; B-G) Attractive → Repulsive	(A-D) Attractive → Repulsive	–	
R102 (BD-368-2)	Attractive/Hydrogen bond → Repulsive	(B-G; A-D; C-J) Attractive/Hydrogen bond → Repulsive	–	
Note:

7CWS does not interact with the viral RBD. The arrow indicates that the mutation changes the interaction type.

Figure 4 Changes in protein structure caused by amino acid substitutions (P.1 strain, P17 antibody, and BD-368-2 antibody).

(A) Altered contacts between the viral spike in the P.1 strain and the main sites from the P17 antibody fragment, with residues labeled, comparing the wild-type spike shown in blue structurally superposed with the mutant model and the antibody in yellow. The colour green represents the earliest interactions, and the mutant contacts are depicted in red. (B) Altered contacts between the viral spike in the P.1 strain and the main sites from the BD-368-2 antibody fragment, with residues labeled, comparing the wild-type spike shown in blue structurally superposed with the mutant model and the antibody in yellow. The color green represents the original interactions, and the mutant contacts are depicted in red. Figure generated using PyMOL.

Alongside, the Beta strain has shown similar, although diminished, disconnection in these points of interest. It is visible in the 7CWU and 7CWN complexes by its correlation in the R96 residue from the P17 antibody, a site where an attractive interaction became constituted of hydrogen bonds. Simultaneously, in the 7CHH complex, attractive interactions in the R100 and R102 residues from the BD-368-2 antibody have altered into repulsive linkages.

As for the Alpha lineage, there were no detected changes in the 7CWU and 7CWN complexes within their interface in the P17 antibody. At the same time, the 7CHH complex has shown differences in the R102 residue from the BD-368-2 antibody, denoted by the transition from attractive interactions and hydrogen bonds to repulsive linkages. The effects of such results for the Alpha and Beta strains have been observed in the study of Wang et al. (2021a, 2021b).

Real-time tracking of COVID-19 B.1.617.2 strain (Delta variant)

The Delta variant has been associated with a viral load approximately 1,260 times greater than the 19A and 19B lineages, becoming more infectious in the early stages (Li et al., 2021) and more transmissible over the Alpha strain (Callaway, 2021). According to the Center for Disease Control and Prevention (CDC), the Delta lineage potentially decreases monoclonal antibody neutralization alongside a reduction in post-vaccination sera in the US. In vaccines that require two shots, Delta has shown to be more infectious (compared to other variants) in people who only had one injection of the compound. Despite this, there were no significant differences in the overall efficacy of the main vaccines (Bernal et al., 2021).

Using the implemented tool E-Volve, we can rapidly find out the electrostatic behavior of a SARS-CoV-2 mutant lineage by only knowing the sequence of the altered spike protein. Utilizing this real-time tracking of protein interfaces, we have perceived key alterations that cause Delta’s clinical characteristics regarding the L452R, P681R, and E484Q spike mutations (Fig. S1).

Like the 6M0J model, the Delta spike protein showed losses of attractive interactions that a priori occurred in the K31/E484 interface. Moreover, it has gained hydrogen bonds between the K31/Q493 and E35/493 complexes. These modifications possibly show us a reconfiguration of the human ACE2 interaction.

Concerning antibody modeled interfaces, within the 7CHH model, the BD-368-2 antibody (7CHH) has shown a major loss of attractive interactions in essential residues such as R100 and R102, probably due to the E484Q mutation. Furthermore, it also has gained highly frequent repulsive interactions in these residues caused by the L452R modification. Subsequently, there have also been alterations such as attractive hydrogen bonds becoming hydrogen bonds in these residues because of E484Q and L452R.

Also, we observed in chain pairs C-M and B-K of the antibody complexes (7CWU/7CWN), that attractive interactions were converted into hydrogen bonds, mainly in the H35, R98, and H99 residues (P17 antibody). Additionally, the results show a frequent loss of attractive connections in these residues, mainly caused by the E484Q mutation. Alongside, the L452R alteration creates a highly repulsive interface involving the K106 residue in these models, further producing hydrogen bonds with the Y32 antibody amino acid.

Molecular dynamics simulations and electrostatic potential calculations

Changes in protein conformational distribution play a pivotal role in the affinity shift in macromolecular complexes (de Lima et al., 2020; Rocha et al., 2021). For the spike-receptor interfaces, a set of studies pointed to the importance of the molecular plasticity for the RBM adequation around the receptor (Carvalho & Alves, 2020; Dehury et al., 2020; Verkhivker, 2020; Ahamad, Hema & Gupta, 2021). Here, to present and discuss our MD results on such aspects, we use the nomenclature from Wang et al. (2020) for the RBD and RBM motifs (Fig. S2). Our molecular dynamics (MD) results show, for wt and P1, higher fluctuations for the C-terminal extension of the loop β1′/β2′ (hereafter called loop β1′/β2′-C and comprising the residues 470 to 490) compared to the rest of the RBD (Figs. 5A and 5B). This is the loop encompassing position 484 close to its center. The higher mobility of the loop β1′/β2-C was recovered both considering its movement relative to the entire RBD as the internal mobility of the RBM (Figs. 5A and 5B, respectively). Also, the β2′/η1′ loop (residues 497 to 506, in whose center the N501Y substitution occurs), as well the loop α1′/β1′ (residues 438 to 450) are local fluctuation hotspots at the RBM and at the RBD as a whole (Figs. 5A and 5B). This system of loops comprises a kind of three-fingered “molecular claw” (the higher finger being the loop β1′/β2′ at one side and the minor being loops β2′/η1′ and α1′/β1′ at the opposite one). This molecular claw “pinches” the borders of the two N-terminal helices from ACE-2 (the H1 and H2 helices) at binding, positioning these helices for deeper contact with the short antiparallel ꞵ1′-ꞵ2′-sheet at the RBM center (Carvalho & Alves, 2020; Wu et al., 2020). In this way, certain plasticity at this region can be expected in order to facilitate this “pinching” movement. Finally, the 417 position (where K417T substitution occurs) is placed at the center of the helix 3,before the RBM and at a less mobile position in Fig. 5A. Although not part of the RBM itself, this position is packed between the C-terminal end of the β1′ strand and the N-terminal part of the loop β1′/β2′ (hereafter called loop β1’/β2’-N) (Figs. 5C and 5D, Table S2). Due to this strategic packing at the center of the RBM, tenuous variations on the packing and mobility of position 417 are likely to influence the fluctuation of the entire domain.

Figure 5 Molecular dynamics (MD) simulation and adaptive Poisson Boltzmann solution (APBS) for wt and Gamma spike’s RBD variants.

(A) Root-mean-square fluctuation (RMSF) of the entire RBD region for wt and Gamma. (B) RMSF obtained aligning the receptor-binding motif (RBM) region and considering only this same region (residues 438-508). The RMSFs are shown as the average (lines) and deviations (shadow) considering the three replicates individually and the sum of the three. The “X” symbol and the residue names mark the three Gamma mutation sites at the RBD contact hotspots depicted in Fig. 1 (K417T, E484K, and N501Y). The red signs are the two mutations placed at the RBM (E484K and N501Y). (C and D) Major cluster conformations for RBM and their percentages recovered at the set of MDs for each one of the respective wt and Gamma lineages colored according to APBS results (from −5 to +5 KT/e in red:white:blue scale). Also, the cluster densities in thousand of frames per cluster average distance (kfr/Å) are shown. The arrow points to the binding site of the H1 helix from ACE-2. The respective K/T417, E/K484, N/Y501, and Y505 residues are highlighted in each case. Figure generated using PyMOL.

A higher fluctuation abrangency along different simulations is observed for the wt RBM than P1, especially at loops where positions 484 and 501 are located (Figs. 5A and 5B, Table S2). It can be noticed that the wt loops β1′/β2′-C and β2′/η1′ reach about 1 Å higher maximum mobility in wt than their respective counterparts in P1 (Figs. 5A and 5B). This reflects a higher distribution of conformational population for the wt RBM at the less dense conformational clusters compared to the Gamma variant, which in turn presents 98.1% of its conformational space centered at the major cluster (Figs. 5C and 5D). Interestingly, the data in Table S2 suggest that this difference seems not to reflect on the RBD and RBM average topology itself, residing the major differences only on the conformational fluctuations around the average. In fact, we can notice that the majority of the weighted average RMSD measures (using the cluster fraction as weights) related to the starting pre-simulation structures (the PDB:6M0J for wt and the homology model based on this PDB for P1) suggest reciprocally similar and not considerable deviations from these initial conformations at the mean, both considering the RBM and RBD, as considering the RBM major motifs (Table S2). However, considering the weighted fluctuations around these same mean values, it can also be noticed a less mobile structure for P1 (for instance, with the average fluctuation of the P1 RMSD of the whole RBM around five times lower than the same fluctuation in wt). The only local exception is a faintly higher fluctuation for the central β1′-β2′ sheet in P1, considering the weighted average measures per cluster representative structures in Table S2. However, the strands of this β-sheet seem to present again lower fluctuations in P1 than in wt when the intra-clusters mobilities are also considered, as can be inferred by the RMSF values in Figs. 5A and 5B.

Only two of the MD recovered measures presented in Table S2 seem to differ the P1 and the wt structures not just in variance, but also considering the average value: the per carbon atom average number of hydrophobic contacts at the 417 side chain (more than one third higher in P1); the average RMSD of the loop α1′/β1 related to the starting structure (more than a half larger in P1). Considering this last issue, the differential context of the α1′/β1 loop at the P1 RBD seems also to be reflected in the fact that this loop is the only part of the P1 RBM with higher RMSF compared to wt (opposite to the behavior of the rest of this motif in Figs. 5A and 5B). Finally, although presenting a less difference in average value, it is also interesting the fact that the mean structure in P1 is stabilized with a 0.1 Å wider RBM than wt (see the higher weighted average Rav radius between the RBM loops in Table S2).

Three major modifications at the P1 internal interactions context seems to contribute to the majority of the topology and mobility distinctions aforementioned between this protein and wt: the higher density of internal positive:positive repulsive interactions (that seems to drive the Gamma RBM more strongly to the more open conformation in Fig. 5D-left side); the respective higher packing of the T417 in helix H3 and Y501 on the center of loop β2′/η1′ with its neighbor residues (specially Y505 at the helix η1′ in the case of Y501) than the analogous K417 and N501 at the wt RBM. These last issues more effectively prevent the adoption of the more “closed” and locally disordered RBM conformations at the Figs. 5C and 5D-right side at the P1 lineage than in wt.

Superposition of the respective conformational centroids for the two major MD clusters for each system with crystal structures, followed by Poisson-Boltzman surface potential calculations, corroborates the higher electrostatic complementarity of the P1 RBM to the ACE-2 interface recovered by E-volve (Figs. 6A–6E). It can be noticed that the ACE-2 region that interfaces with the spike RBM has a highly negative Poisson-Boltzmann potential surface (Fig. 6A), besides a high negative electrostatic potential at its intracellular domain as a whole (not shown). In this way, considering just the more populated “all open” conformation (more similar to the crystal-like conformations analyzed at the E-Volve procedures), we can already note a minor electrostatic complementarity between the wt RBM and this receptor, due to the local negative “patches” of wt, than for the more widely positive Gamma RBM (left side structures on Figs. 6B–6E).

Figure 6 Superposition of the MD major cluster’s conformations for the RBD of the wt and P.1 lineages with the ACE2 interface of the PDB:6M0J colored according to APBS results.

(A) ACE-2 interface with spike RBM (ACE2 interph.) at PDB:6M0J, highlighting the helixes H1 and H2 regions. (B–E) Respective conformations and percentages for the two major clusters recovered at the wt and P.1 MDs superposed at the ACE2 interph. (in transparent) from PDB:60MJ, shown at two orientations each (top and bottom). The spike RBM (S-RBM) is identified in B. In C, the two highly negative regions, locally repulsive to the ACE-2 contact, emerging at the second most populous wt cluster (8.4% of the conformations for this lineage) are surrounded by dashed circles. The arrows point to the respective E/K484 residues in wt and P.1 structures. Figure generated using PyMOL.

A similar behavior but with opposite consequences for the differential affinity to wt and Gamma lineage, is observed when comparing the stereochemical/electrostatics complementarity of the respective MD major clusters to the P17 and BD-368-2 interfaces (Fig. 7). It can be noticed that the higher density of negative subsites at the RBM region for the wt conformers (accentuated at the second cluster) turns this protein substantially more complementary to the positive interface of the two antibodies (overall P17) than the widely positive Gamma RBM, again in accordance with the single structure analysis by E-volve.

Figure 7 Superposition of the MD major cluster’s conformations for the RBD of the wt and P.1 lineages with the P17 and BD-368-2 antibodies from the respective PDBs 7CWN and 7CHH colored according to APBS results.

(A and B) P17 and BD-368-2 respective spike interfaces (A. inter.). The antibodies (Ab) are colored according to APBS, while the spike RBD is in white color and transparent. Dashed geometric forms depict the higher Ab-RBD contact points. (C and D) The two wt’s MD major cluster centroids bound to P17 and BD-368-2. All the proteins are colored according to APBS and Ab are transparent. Dashed circles depict regions with significant charge-to-charge interprotein attractive contacts (overall at the secondary cluster conformer with 8.4% representation) compared to the P.1-Ab complex. (E and F) Same image considering P1’s MD major cluster superposition to the antibody complexes. Figure generated using PyMOL.

A complement that the MD conformational search brings to the previous E-volve single structure analysis, however, is that the aforementioned alteration at the conformational fluctuation between both lineages can reinforce the differential affinity to ACE-2 and antibodies (Figs. 6 and 7). The lower electrostatic complementarity to ACE-2 seems to be still accentuated at the second cluster of the wt RBM (adopted by this protein with 8.4% statistics along the MD set) due to the movements at the RBM loops, which result in a higher exposition of the negative “patches” at its interfacial region (right side structures at Fig. 6B and 6D). Although the local alterations of the Gamma’s RBM second major cluster seem to result in a minor contact surface area with ACE-2, two crucial issues diminish the implications of this conformation on attenuation of the spike-receptor affinity. First, there is no apparent emergence of probable charge to charge repulsion with the ACE-2 surface from the first to the second conformation in Gamma. Second, the less complementary second cluster conformation only appears in 1.5% of the MD simulation, suggesting a lower contribution for this conformation at the RBM phase space than for its analogous in wt (right side structures in Figs. 6C and 6E).

Taking all together, the structural and electrostatic analysis for the major conformers recovered by clustering procedure at equilibrium MD simulations corroborate the general inference from E-Volve inspections. In both cases, it is recovered a higher complementarity to the human ACE2 receptor and a lower one to known antibody lineages for the P1 mutant than for the wt RBD considering the most populous conformers in MD, or the single representative conformer previously taken to E-volve. In addition, the MD and APBS analysis also points out some new enhancing features of such effects when the protein conformational equilibrium is considered.

The higher disorder of the loop α1′/β1′ at the P1 RBM suggests an additional impedance for antibody binding

As previously mentioned, the principal subregions in which interactions with the receptor and/or antibodies occur and/or change at the static structures analyzed with E-volve in Figs. 2–4 (i.e., the β1′-β2′ sheet, the loops β1′/β2′-C and β2′/η1′) maintain an average topology equally similar to these starting conformations along the MDs (Table S2). The same can be said about the average structure of the entire RBM and the RBD as a whole. In this way, the major conclusions previously drawn from the E-volve analysis about the nature of the major interactions around the major part of the RBM/RBD sites are maintained. It seems to be only necessary, for a more accurate discussion, to incorporate to these same conclusions the enhancing effects carried by the distinct conformational fluctuations for wt and P1 RBMs around the average. However, a single specific RBM subsite presenting peripheral interactions with ACE-2 and antibodies, the loop α1′/β1′ (residues 438 to 450), seems to suffer a change concerning the average structure in P1 compared to wt (Table S2). The differential context of this loop is still corroborated by its higher RMSF in P1 compared to the same region in wt, a behavior in contrast to the recovered for the rest of the RBM. In this way, a more detailed analysis of the differences in conformational space for this only loop at the wt and P1 respective contexts, as well their possible implications for binding, seems to be necessary.

Superpositions of representative RBM structures from our MD data with the protein:protein interfaces at different complexes are depicted in Fig. S3. The superpositions are carried, from right to the left, for the RBM binding interfaces with ACE-2, the antibody P17 and antibody BD-368-2, as well as at the context of the starting (crystal-like), structures (Fig. S3, top row), the three cluster centroids from wt MD simulations (Fig. S3, middle row) and cluster centroids from P1 MD simulations (Fig. S3, bottom row). It can be noticed that both the higher average RMSD related to the pre-MD structure and the higher RMSF at the P1 loop α1′/β1′ seem to be associated with a local less structure of the helix α1′ at its basis. This leads to a higher disorder and mobility of the loop as a whole (Fig. S3, bottom row). This less ordered loop α1′/β1′ in P1 leads to representative clusters in which the topology of the same loop is more dissimilar to the binding conformation than in wt. Once the binding conformation of this loop implicates on its higher ordering, it is expected a greater entropic penalty to this organization for the naturally disordered loop in P1 than in wt (naturally ordered) (Batori, Koide & Koide, 2002; Benfield et al., 2006; Rajasekaran et al., 2016). In addition, an apparent structural clash of the more occupied structures for the binding at the BD-368-2 antibody, overall considering the residue Y449, seems to be expected for the P1 RBM, but not for wt (Fig. S3, last column). Although more specific studies about this conformational and dynamics influences at complexation need to be carried out (studies already in development in our group), the binding modes suggest that the penalties above mentioned would be lower at ACE-2 association, middle at P17 and higher at the BD-368-2 antibody complex. At the ACE-2 complexation, the residues at the RBM loop α1′/β1′ just interact with a small fraction of the receptor interfaces at the C-terminal extension of the ACE-2 helix H1. A major part of the ACE-2 interaction is mediated by the protruding of the middle and N-terminal portions of H1 at the RBM β1′-β2′ sheet, loops β1′/β2′ and β2′/η1′ region (dashed arc at Fig. S3, right and bottom). In this way, it is expected that the higher fit between the RBM and the ACE-2 interfaces at these regions by all the aforementioned effects compensates sufficiently for the conformational difficulties for the loop α1′/β1′ interactions. Considering the antibody P17, two respective cavities at this antibody mediates the respective interactions with the RBM loops β1′/β2′ and α1′/β1′ (dashed arcs in Fig. S3, bottom row, middle column). Although the importance of the interaction with loop α1′/β1′ has increased compared to the complex with ACE-2, the interaction with the loop β1′/β2′ are still higher, in such a way that the findings at E-volve analysis can, still, be of more importance to describe the loss of affinity. For the complex with the BD-368-2 antibody, however, a reorientation of the antibody at the protein:protein interface compared to P17 seems to put the interaction with the loop α1′/β1′ as being of greater relevance for the complex stability as a whole (Fig. S3, bottom row, right column). In this way, we advocate that the higher disorder of the loop α1′/β1′ at the P1 mutant can be a specific evolutive strategy to draw back interactions with antibodies with this similar interaction mode. Although this dependence of subtle conformational changes at specific complexes interactions and stability was not taken into account in our case study of the E-volve applications, it is entirely possible to carry previous short MD studies and collection of the more populated conformers before E-volve applications, what can be of more reliable finds for future studies of groups interested on to measure the impact of new spike mutations.

Limitations and perspectives

One current limitation of E-Volve is that the models do not present the conformational changes of the generated proteins. The models created in E-Volve are generated by the MODELLER software, which has limitations in the homology modelling algorithm. The output model does not show the conformational changes that occur in the real ligands. Furthermore, mutations that could alter the spike stereochemical configuration, even in regions outside of the analyzed locus, will not be foreseen by E-Volve. Despite this, we believe E-Volve is significant to real-time tracking of structural mutations in new and emerging strains, until the experimental structure is available. It is also interesting to highlight that in our case studies, the major structural features of the MODELLER recovered models have maintained themselves at the average structures recovered at the MD protocols, just being necessary to add to the final conclusions the effects of the distinct structural fluctuations around these averages. This is not an unexpected finding, once that usually, a small number of single amino acid substitutions on the surface of a protein result in a final molecule with basically the same global folding, but changing local attributes related to interactions and mobility (Wang & Moult, 2001). Anyway, it is possible to increment in future a version in which, after the generation of the models by MODELLER, a short conformational search, for example, using the simplified Monte Carlos based protocols from the ROSETTA software (Brunette & Brock, 2008; Potrzebowski, Trewhella & Andre, 2018) is carried, taken the most probable conformer(s) to the posterior docking procedures. It is also possible to incorporate (once a server with sufficient computing power is available) previous short molecular dynamics simulation protocols, overall for the RBM loops, after the modeling and before docking. Finally, it is equally possible, in future, to implement new versions of E-Volve with a set of other molecular modeling algorithms in which eventual small conformational changes can be more accurately considered. By now, the experimental structures of Alpha, Beta and Gamma are available on PDB.

Conclusion

In this paper, we introduced the E-Volve web tool. As proof of concept, we have evaluated the impact caused by mutations on the spike’s RBD as for the non-covalent interactions altered between the spike-ACE2 and the spike-Antibody complexes. In addition, our research scrutinized an interactional increase in highly immunogenic regions in the RBD of the SARS-CoV-2. Those mainly related to the 501, 484, and 417 amino acid positions are also frequent in most models studied. Furthermore, the implementation of E-Volve guides users, in an automatized way, to evaluate and comprehend the process we scrutinized in this research. The tool is useful for constant analysis of the data generated by the real-time tracking of the COVID-19 pandemic. E-Volve facilitates the comprehension of SARS-CoV-2 emerging mutant lineages, rapidly contributing to worldwide epidemiology. Moreover, the tool creates a useful intelligible environment for general protein mutation analysis and evaluation, as demonstrated in this study. Once mutations with higher potential impact to the receptor or immune system interactions are firstly prospected by this tool (on an extensive list), deeper analysis can be individually carried out just for these hits in individuals (as exemplified by our MD comparative studies with the wt and Gamma lineages). This may represent a substantial gain in time for research for new impacting variants. E-Volve encompasses biological systems with amino acid residue alterations, from minor to major scale structures, depicting quality-based bioinformatics models in protein mutagenesis and poses an extending step for proteomics studies.

Supplemental Information

Supplemental Information 1 Supplementary Figures and Tables.

Figure S1. Inter-chain interactions in spike protein for Delta (PDB ID 7V8B) and Omicron (PDB ID 7T9K) variants.

Figure S2. MD results for the RBD and RBM motifs.

Figure S3. Influence of the differential context of the loop α1′/β1′ on the complementarity of the WT and P1 RBDs at the receptor and antibodies.

Table S1. List of 3D-structures (PDB IDs) collected from Protein Data Bank (https://www.rcsb.org/). Data was collected on April 30, 2021.

Table S2. RMSD concerning the starting structure, packing and mobility parameters recovered at the respective MD simulations for the WT and P1 RBD.

Click here for additional data file.

Additional Information and Declarations

Competing Interests

Author Contributions

Data Availability

The authors declare that they have no competing interests.

Vitor Pimentel Dos Santos performed the experiments, authored or reviewed drafts of the paper, and approved the final draft.

André Rodrigues analyzed the data, prepared figures and/or tables, authored or reviewed drafts of the paper, and approved the final draft.

Gabriel Dutra performed the experiments, analyzed the data, authored or reviewed drafts of the paper, and approved the final draft.

Luana Bastos performed the experiments, authored or reviewed drafts of the paper, and approved the final draft.

Diego Mariano performed the experiments, prepared figures and/or tables, authored or reviewed drafts of the paper, and approved the final draft.

José Gutembergue Mendonça performed the experiments, analyzed the data, authored or reviewed drafts of the paper, and approved the final draft.

Yan Jerônimo Gomes Lobo performed the experiments, analyzed the data, authored or reviewed drafts of the paper, and approved the final draft.

Eduardo Mendes analyzed the data, authored or reviewed drafts of the paper, and approved the final draft.

Giovana Maia analyzed the data, authored or reviewed drafts of the paper, and approved the final draft.

Karina dos Santos Machado analyzed the data, authored or reviewed drafts of the paper, and approved the final draft.

Adriano Velasque Werhli analyzed the data, authored or reviewed drafts of the paper, and approved the final draft.

Gerd Rocha performed the experiments, analyzed the data, authored or reviewed drafts of the paper, and approved the final draft.

Leonardo Henrique França de Lima conceived and designed the experiments, performed the experiments, analyzed the data, authored or reviewed drafts of the paper, and approved the final draft.

Raquel de Melo-Minardi conceived and designed the experiments, performed the experiments, analyzed the data, prepared figures and/or tables, authored or reviewed drafts of the paper, and approved the final draft.

The following information was supplied regarding data availability:

The datasets presented in this study can be found at VTR: http://bioinfo.dcc.ufmg.br/vtr/covid_home.

The source code of scripts used by the E-Volve web tool is available at GitHub: https://github.com/Dutra161/E-volve.

The user-friendly web tool is available at E-Volve: http://bioinfo.dcc.ufmg.br/evolve.

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
