# Peer review of "E-Volve: understanding the impact of mutations in SARS-CoV-2 variants spike protein on antibodies and ACE2 affinity through patterns of chemical interactions at protein interfaces"

_PeerJ, doi:10.7717/peerj.13099_

## Round 0.1 · original submission · Major Revisions

Authors are not required to agree with every suggestion made by the reviewers. However, reasons for opposing any reviewer’s comments must be provided.

Please pay close attention to the section Validity of the findings written by Reviewer 1, the Major issue raised by reviewer 2, the model validation pointed by Reviewer 4, and a better description of the statistical methods used to validate your results.

Two reviewers have also raised concerns about the English proofreading of your manuscript. Thus, I would recommend that your text should be revised by a proficient English speaker before the new submission.

Reviewer 1 ·

Basic reporting

The English language in the manuscript should be improved to ensure that your research is clearly understood and appreciated by the international audience. I have listed specific details in my comments below but overall, the introduction, result, and conclusion sections need major rewriting due to the use of ambiguous English language. I suggest the author contact a professional editing service to assist them. This will add great value to this research manuscript.

Detailed comments:
Authors should provide an Abstract which is unified and concisely reports the research problem, methods, key results, and conclusion. Currently, authors have segmented each of these sections as a substitute for a more succinct and clearer abstract.

Please reframe the paragraph starting from lines 27-32 Page 7 it's not clear and difficult to follow for the reader.

Line 35-36 Please clarify what do you mean by electrical alterations? Please re-frame.

Line 40 page 7 "this effect by changes at the mutant’s dynamics equilibrium" authors should re-write this paragraph as it's unclear what the author meant by “mutant’s dynamics equilibrium”.

The use of the word "empirically" by the authors doesn’t seem relevant. It is often used for citing other researchers' work in the manuscript. Authors should use “as observed” or provide more details as to why they think other cited studies are empirical in nature. One such example is Line 376, page 16 where authors are citing work by Wang et al. 2021.
I thank the authors for providing relevant detailed figures however in the case of the delta variant protein-protein interaction authors should provide a figure similar to Figure 4 as supplementary material.

Please place the references in order as they appear in the manuscript. Example: Line 49 page 8 reference (Hu et al. 2021) is mentioned but can't be found in the reference list.
Please provide a reference for Line 26 page 7 written in the Abstract
Page 8 Line 56-59 Please provide relevant references for this statement.
Line 91 page 8 Authors should use the full form of ACE2 (Angiotensin-Converting Enzyme 2) at the earlier occurrence of ACE2 in the manuscript (introduction section).

Line 413 page 17 special characters "฀?1'/?฀2’loop" are not displayed properly in the text. Please correct this across the entire manuscript

Line 109 page 9 Supplementary Table S1 provided by the authors doesn’t provide any descriptions for the list of PDB's, please provide a description for these PDBs.

Authors should choose one nomenclature for the mutants like alpha, beta, delta and consistently use that in the manuscripts rather than switching between nomenclatures B1.1.X vs Greek names of the variants.
Please provide high-quality 300 DPI images to be included in the manuscript. Example: The set of images in Figure 1 are hard to read, and gets blurred when zoomed in.

Experimental design

Line 116 page 9 please provide a reference for the MODELLER methodology or describe it briefly in the manuscript’s Material and Method section.

Line 274-280 Page 13 Authors should elaborate on the TM-score and describe the methodology they have employed to generate the heat maps. For example, it was not clear how a “previously generated database” (constituents of this database are not detailed) is employed to emphasize the impact of mutations on protein-protein interactions?

Validity of the findings

Line 333 Page 15 "Y501/Y41 bond, hydrophobic interaction in the spike-ACE2 complex" contradicts with Figure 3B where Y501/Y14 interaction is shown. Authors should clarify which interaction is correct Y501/Y14 or Y501/Y41 and make changes either in the text or figure.

In Figure 5A mobility around the 417 position doesn’t seem to be significant compared to nearby positions, authors should clarify or provide more evidence to support their statement made regarding the mobility around 417 positions (Line 421 Page 17)

I would like the authors to comment on the higher mobility observed around the 438-450 residues for the P1 (gamma mutant) compared to wildtype (Figure 5B) and its implication on the conformational state of the protein.

I would recommend that the authors report only the observed results in the Result section of the paper and elaborate their findings in more detail in the discussion section. Currently, it’s not the case making it difficult for the readers to focus on the key findings.

Additional comments

In this manuscript the authors have demonstrated a web-based tool named "E-Volve", to systematically analyze interchain interactions given a PDB identifier, list of mutations, chain pairs as inputs. The tool evaluates protein-protein interaction using a contact comparison strategy. Authors have used E-Volve to analyze variants of SARS-CoV-2 spike proteins and their impact on their ability to interact with ACE2 and antibodies. I commend the authors for their extensive work on the SARS-COV-2 spike's protein-protein interactions. I appreciate the use of molecular dynamics simulations for the gamma variant studies that corroborate the findings from their proposed E-Volve tool. However, the manuscript needs to be re-written in an unambiguous language before acceptance for publication

·

Basic reporting

Please see my additional comments.

Experimental design

Please see my additional comments.

Validity of the findings

Please see my additional comments.

Additional comments

The manuscript entitled “The E-Volve: understanding the impact of mutations in SARS-CoV-2 variants spike protein on antibodies and ACE2 affinity through patterns of chemical interactions at protein interfaces” by
Vitor Pimentel Dos Santos et al. investigated the mechanisms that mutations in the viral spike protein of SARS-CoV-2 have decreased the activity of neutralizing antibodies, and increased the affinity to ACE2 receptor using E-Volve. Authors have demonstrated highly frequent interface changes mediated by mutations (N501Y and E484K). The spike-ACE2 complex induced new hydrophobic bonds. Remarkable electrical alterations that breached important sites in the spike-Antibody complexes were observed. This study is of clinical significance, providing a web tool for real-time tracking of the virus mutant lineages. It will potentially help doctors adjust treatments according to virus mutations. However, some issues need solving.
Major issue:
1. This study manuscript were largely dependent on E-Volve algorithm. How do authors validate their findings experimentally?
Minor issues:
2. Why was Figure 1C referred before Figure 1B?
3. In text line 412-421, fonts are not consistent and there are some typos.

Reviewer 4 ·

Basic reporting

no comment

Experimental design

Various methods needs to be used to validate the models.

Validity of the findings

no comment

Annotated reviews are not available for download in order to protect the identity of reviewers who chose to remain anonymous.

---

## Round 0.2 · accepted · Accept

Your revised manuscript was accepted by one of the reviewers that previously asked for a revision.

·

Basic reporting

Authors have successfully responded to my comments.

Experimental design

None.

Validity of the findings

None.

Additional comments

None.